# The Human Myofibroblast Marker Xylosyltransferase-I: A New Indicator for Macrophage Polarization

**DOI:** 10.3390/biomedicines10112869

**Published:** 2022-11-09

**Authors:** Thanh-Diep Ly, Monika Wolny, Christopher Lindenkamp, Ingvild Birschmann, Doris Hendig, Cornelius Knabbe, Isabel Faust-Hinse

**Affiliations:** Institut für Laboratoriums- und Transfusionsmedizin, Herz- und Diabeteszentrum NRW, Universitätsklinik der Ruhr-Universität Bochum, Georgstraße 11, 32545 Bad Oeynhausen, Germany

**Keywords:** cytokines, fibrosis, inflammation, macrophage, proteoglycan, systemic sclerosis, xylosyltransferase

## Abstract

Chronic inflammation and excessive synthesis of extracellular matrix components, such as proteoglycans (PG), by fibroblast- or macrophage-derived myofibroblasts are the hallmarks of fibrotic diseases, including systemic sclerosis (SSc). Human xylosyltransferase-I (XT-I), which is encoded by the gene *XYLT1*, is the key enzyme that is involved in PG biosynthesis. Increased cellular *XYLT1* expression and serum XT-I activity were measured in SSc. Nothing is known so far about the regulation of XT-I in immune cells, and their contribution to the increase in measurable serum XT-I activity. We utilized an in vitro model, with primary human CD14^+^CD16^+^ monocyte-derived macrophages (MΦ), in order to investigate the role of macrophage polarization on XT-I regulation. The MΦ generated were polarized towards two macrophage phenotypes that were associated with SSc, which were classified as classical pro-inflammatory (M1-like), and alternative pro-fibrotic (M2-like) MΦ. The fully characterized M1- and M2-like MΦ cultures showed differential XT-I gene and protein expressions. The fibrotic M2-like MΦ cultures exhibited higher XT-I secretion, as well as increased expression of myofibroblast marker α-smooth muscle actin, indicating the onset of macrophage-to-myofibroblast transition (MMT). Thus, we identified XT-I as a novel macrophage polarization marker for in vitro generated M1- and M2-like MΦ subtypes, and broadened the view of XT-I as a myofibroblast marker in the process of MMT.

## 1. Introduction

The term ‘tissue fibrosis’ summarizes a pathological condition that is characterized by the excessive expression and accumulation of extracellular matrix (ECM), leading to organ dysfunction. It plays a role in the progression of many chronic diseases, such as systemic sclerosis (SSc), and is not restricted to certain organs [1,2]. The generation of α-smooth muscle actin (α-SMA)-positive cells, called myofibroblasts, with high ECM-synthesizing capacities, is considered to be a key step in fibrosis development [3]. Although classically thought to be derived from tissue-derived fibroblasts, myofibroblast progenitor cells include circulating fibrocytes/monocytes from bone marrow and endothelial or epithelial cells [4]. Lately, the transition process of macrophage to myofibroblasts (MMT) has been implicated in fibrotic diseases, such as renal or lung fibrosis [5].

Circulating human monocytes expressing cluster of differentiation (CD)14 and/or CD16 are the largest type of peripheral blood mononuclear cells (PBMC). According to their CD14/CD16 surface expression ratios, monocytes can be subdivided into classical (85%), intermediate (5%) and nonclassical (10%) monocytes. All monocyte subtypes can migrate into the tissue and differentiate into macrophages as part of the immune response [6,7]. In addition to their phagocytic activity, which is vital for the host defense against microorganism or apoptotic cell clearance, macrophages contribute to the remodeling of the ECM by sensing tissue damage and facilitating tissue repair responses after injury. Consequently, abnormal macrophage function or activation contributes to pathological conditions, such as fibrosis [8]. The homeostatic control of the macrophage pool is mediated by the growth factor macrophage colony-stimulating factor (M-CSF), which is produced by several cell types, including macrophages and fibroblasts; meanwhile, granulocyte-macrophage colony-stimulating factor (GM-CSF) drives macrophage differentiation during inflammatory responses, as GM-CSF secretion by the same cell types requires inflammatory stimuli to be expressed [9]. Therefore, the in vitro generation of monocyte-derived macrophages (MΦ) as a model for tissue macrophages can be obtained using PBMC-derived monocytes, with either M- or GM-CSF. The M-CSF-matured MΦ (M-MΦ) or GM-CSF-matured MΦ (GM-MΦ) can respond to their microenvironment by obtaining a certain type of inflammatory state designated as naïve, unpolarized (M0), pro-inflammatory, classical-activated (M1-like) or anti-inflammatory, profibrotic, alternatively activated (M2-like); this state is known as macrophage polarization [10,11]. M1-like MΦ are thought to promote tissue damage by the secretion of inflammatory cytokines for effective host defense against intracellular pathogens. Thus, M1-like MΦ can be obtained in vitro by the stimulation of M0 M- or GM-MΦ, with T helper (T_H_) 1 cell cytokine interferon-γ (IFN-γ) and Toll-like receptor ligand lipopolysaccharide (LPS). By contrast, M2-like MΦ promote tissue regeneration after inflammation and/or injury by producing fewer pro-inflammatory cytokines, while increasing the secretion of ECM components, such as heparan sulfate (HS) proteoglycans (PG). Therefore, M2-like MΦ can be generated in vitro by the stimulation of M0 M-MΦ with T_H_2 cytokine interleukin (IL)-4 [12,13,14]. Common cytokine gene and surface markers that are used to distinguish human M1- and M2-like MΦ subtypes are the pro-inflammatory cytokines IL-1β, IL-8 and IL-6, as well as class II major histocompatibility complex (MHC II; human leukocyte antigen-DR isotype, HLA-DR) and CD80, which promote cytotoxic adaptive immunity in M1-like MΦ. In addition, transforming growth factor (TGF)-β, IL-10, hemoglobin scavenger receptor (CD163) and mannose receptor I (CD206) are commonly used to discriminate the M2-like phenotype. When using IL-4 for M2-like MΦ generation, this phenotype is designated as M2a, whereas other M2-like phenotypes exist, such as M2c. Macrophages with this classification are associated with good wound healing properties, expressing increased IL-10, TGF-β1 and CD163 levels. This subtype can be generated in vitro by MΦ stimulation with IL-10 or glucocorticoids [11,15,16].

As macrophages are known to constitutively express low levels of pro-inflammatory cytokine IFN-γ, the bioavailability and action of this cytokine on macrophage polarization is strictly controlled by cell surface heparan sulfate proteoglycans (HSPG), which is the major PG type that is produced by macrophages, and vice versa [10,17]. The HSPG are macromolecular structures present on the cell surface or in the ECM. The PG are structurally characterized by a PG core protein that is post-translationally modified with linear polysaccharide chains, the glycosaminoglycans (GAG). The GAG, in turn, are made of repetitive disaccharides that are highly sulfated. The biological role and classifications of PG are based on their GAG, which include, inter alia, chondroitin sulfate (CS) and HS [18]. As such, transmembrane HSPG syndecan (SDC) or secreted perlecan (HSPG2) exhibits multiple roles in inflammatory responses, including cytokine and growth factor activity, cell migration and maturation. Thus, the profile of PG expression is modified during monocyte to macrophage transition, or during macrophage polarization [14,17]. PG dysfunction, therefore, has major pathological implications, such as arthritis, diabetes, neurodegenerative diseases and atherosclerosis [19].

The human xylosyltransferase (XT) is the first enzyme of an enzyme series that consists of glycosyltransferases, epimerases and sulfotransferases, which mediates GAG assembly. Human XT catalyzes the transfer of the first monosaccharide residue xylose to specific serine residues of the PG core protein, which is the rate-limiting step for GAG chain elongation in PG biosynthesis. Two XT isoforms (XT-I, -II), which are encoded by the genes *XYLT1* and *XYLT2*, are implicated in human PG biosynthesis. Mutations in human *XYLT1* cause rare diseases with skeletal abnormalities [20]. Various mediators, such as the TGF-β family cytokines TGF-β1 and activin A, IL-1β or LPS, have been shown to regulate *XYLT1* expression and XT-I activity in non-immune cells, but not that of human XT-II isoform [21,22,23]. The *XYLT1* regulatory action of the mediators were shown to be mediated by the regulation of canonical or non-canonical components of the TGF-β pathway [24]. Although XT-I is a transmembrane protein that is localized in the endoplasmic reticulum and/or cis-Golgi compartment, the majority of cellular XT-I activity was quantified extracellularly [25,26]. Thus, the increased serum activity of XT-I is associated with a high rate of PG biosynthesis, and has further significance in fibroproliferative diseases such as liver, renal or dermal fibrosis [19,21,27,28]. Thus, the increased myofibroblast content is marked by increased α-SMA and XT-I expressions in human primary fibroblast cultures. Therefore, the enzyme activity of human XT-I is not only an in vivo serum biomarker for fibrotic remodeling processes, but XT-I expression represents a genotypic and enzymatic marker for in vitro differentiated myofibroblasts [21,29].

To the best of our knowledge, nothing is known so far about the regulation of XT-I activity in immune cells, such as macrophages, and their contribution to the changes in measurable serum XT-I activity. The aim of this study was to explore the expression patterns of the former postulated myofibroblast marker XT-I and its isoform XT-II in human M1- and M2-like MΦ, and their suitability as novel marker genes to distinguish human MΦ subtypes. The results of this study provide a first key insight into XT-I regulation in immune cells and, furthermore, improves the understanding of macrophage biology.

## 2. Materials and Methods

### 2.1. Materials

The materials, reagents and solutions that were necessary for the negative selection of monocytes were obtained from Miltenyi Biotec (Bergisch Gladbach, Germany). Such materials included the PAN monocyte isolation kit, the MACS BSA stock solution and autoMACS rinsing solution, the MACS separator and the MACS column LS. The recombinant cytokines and growth factors M-/GM-CSF, IL-4 and IFN-γ were obtained from PeproTec (Hamburg, Germany). The LPS from the *E. coli* serotype O111:B4 was obtained from Sigma-Aldrich (St. Louis, MO, USA). The Lipofectamine 2000 (LF2000) transfection reagent, 1× Opti-MEM I serum reduced media (OptiMEM), and the *XYLT1* targeting siRNA (ID 112165) and non-targeting siRNA (silencer negative control #1) for cell transfection were purchased from Thermo Fisher Scientific (Waltham, MA, USA).

### 2.2. Human Samples

Buffy coats, with an approximate volume of 50 mL from healthy anonymous donors, were obtained from the blood department of the Institute for Laboratory and Transfusion Medicine, Heart and Diabetes Center (HDZ) NRW, University Hospital of the Ruhr University Bochum (Bad Oeynhausen, Germany). Informed consent was waived due to the usage of anonymous donor samples. The study was approved by the Ethics Committee of the HDZ NRW, Department of Medicine, Ruhr University of Bochum (registry no. 2022-916, 07 March 2022).

### 2.3. Isolation and Cryopreservation of PBMC from Buffy Coats

The isolation of fresh PBMC from buffy coats was performed using density gradient centrifugation with Cytiva Ficoll-Paque PLUS medium (1.078 g/mL; Thermo Fisher Scientific, Waltham, MA, USA) under standardized aseptic conditions. An amount of 50 mL buffy coat was diluted to a 1:2 volume ratio with prewarmed (37 °C) Dulbecco’s phosphate buffered saline (1× PBS; Thermo Fisher Scientific, Waltham, MA, USA). A volume of 25 mL of the buffy coat dilution was layered on top of 25 mL of density gradient medium. A total of four conical 50 mL centrifugation tubes were prepared per initial buffy coat sample. After a centrifugation step (760× *g*, 20 min, brake off), the mononuclear cells were obtained by removing the upper layer, and collecting the cells from the interface in separate tubes. The cells were washed three times (350× *g*, 8 min, brake on) with pre-warmed wash medium (1× PBS supplemented with 10% [*v*/*v*] Roswell Park Memorial Institute [RPMI]-1640 medium [Thermo Fisher Scientific, Waltham, MA, USA]) that was added to a final volume of 50 mL. An additional washing step at a lower centrifugation speed (200× *g*, 10 min, brake on) was performed with the combined mononuclear cells of two centrifugation tubes, in order to remove residual thrombocytes. The cells were diluted in precooled (4 °C) Red Cell Lysis Buffer (10 mL; PromoCell, Heidelberg, Germany) and incubated for 15 min at room temperature (rt), in order to lyse residual erythrocytes. Thereafter, prewarmed wash medium was added to a final volume of 50 mL, followed by a centrifugation step (200× *g*, 10 min, brake on). The cells were combined and diluted with wash medium for the differential blood count, using the Sysmex XN-1000 hematology analyzer (Sysmex Corporation, Kobe, Japan). After centrifugation (200× *g*, 10 min, brake on), the cells were resuspended to yield a final white blood cell (WBC) concentration of 1 × 10^8^ WBC/mL, using complete RPMI-1640 medium (10% [*v*/*v*] fetal bovine serum [FBS; Biowest, Nuaillé, France], 2 mM L-glutamine [PAN Biotech, Aidenbach, Germany], 1% [*v*/*v*] Penicillin-Streptomycin-Amphotericin Mix [100×; PAN Biotech]). A total of 5 × 10^7^ WBCs per cryotube were prepared by adding 50% (*v*/*v*) of the cell suspension to 50% (*v*/*v*) freezing medium (20% [*v*/*v*] dimethyl sulfoxide [Carl Roth, Karlsruhe, Germany] in FBS). The prepared cryotubes were slowly cooled down to −80 °C within 24 h, before being stored in liquid nitrogen.

### 2.4. Generation and Polarization of Human Primary MΦ

The isolation of untouched monocytes from cryopreserved human PBMCs was performed using the PAN Monocyte Isolation Kit, according to the manufacturer’s instructions. The magnetic separation using the LS column was conducted with a maximum total of 1.5 × 10^8^ WBCs per column, which correspond to the generation of three of the cryotubes. Upon purification, the cell suspension was examined for monocyte content using the FCS/SSC plot of an automated hematology analyzer (Sysmex XN-1000, Kobe, Japan). All cell suspensions with a monocyte content above 80% were used for downstream procedures.

Regarding macrophage generation, the negatively selected monocytes were grown under standardized aseptic culture conditions (37 °C, 5% CO_2_) on tissue culture-treated substrates in the presence of M-CSF (50 ng/mL) or GM-CSF (50 ng/mL), which were supplemented with complete RPMI-1640 medium for 6 to 8 days. The day of monocyte seeding was considered to be day 0. A change in medium was performed on day 3 by replacing 50% (*v*/*v*) of the cell culture supernatant with fresh complete RPMI-1640 medium that was supplemented with twice the growth factor concentration. The monocyte number and growth media volume utilized depended on the assay being conducted. A total of 2 × 10^6^ monocytes per 60 mm dish was grown, with 4 mL medium for flow cytometry and cellular XT-I activity determination. The qRT-PCR analysis and IL-6 determination were conducted with 1 × 10^6^ initial monocytes per well of a 6-well plate, using 2 mL of medium. Removable 8-well chamber glass slides (Ibidi, Gräfelfing, Germany) with 1 × 10^5^ cells per well and 0.4 mL of medium were prepared for the immunofluorescence analysis. On day 6, the generated macrophages were polarized towards an M0, M1- or M2-like phenotype by replacing the complete cell culture supernatant with fresh growth medium that was supplemented with adequate cytokines. The volume of polarization medium used equaled that of the cell cultivation, except for the determination of the extracellular XT-I activity, which was performed with half of the volume. The generation of M1-like M- or GM-MΦ was achieved by adding 50 ng/mL IFN-γ and 10 ng/mL LPS to the M-CSF- or GM-CSF-supplemented complete RPMI-1640 medium, respectively. M2-like M-MΦ were generated by the addition of 20 ng/mL IL-4 to the M-CSF-supplemented growth medium. The MΦ that were cultured with solely M-CSF- or GM-CSF-supplemented complete RPMI-1640 medium resembled M0 M- or GM-MΦ. Cells were polarized for 24 or 48 h, and used on day 7 or 8 for downstream analysis.

### 2.5. Transfection of Human Primary Monocyte-Derived Macrophages

The forward transfection of attached primary human M-MΦ was performed on day 5. The cell monolayer was prepared for the transfection procedure by washing with prewarmed 1× PBS to remove floating cells and antibiotic residues, and by adding fresh RPMI-1640 media constituted with 10% (*v*/*v*) FBS, 2 mM L-glutamine and 75 ng/mL M-CSF. The media volume that was added per well depended on the plate format utilized: 1 mL per well of a 6-well plate, or 2 mL for 60 mm cell culture dishes.

An siRNA- and a LF2000-OptiMEM mixture were composed to prepare an siRNA-LF2000 complex for each transfection sample. A 5 μM siRNA working solution in RNAse-free water (Thermo Fisher Scientific, Waltham, MA, USA) was used for the siRNA-OptiMEM mixture preparation. The required volume of siRNA working solution was diluted in OptiMEM to yield a final siRNA concentration of 50 nM per well. The LF2000-OptiMEM mixture was prepared by diluting LF2000 to a 1:50 volume ratio with prewarmed OptiMEM. After incubation for 5 min at rt, the LF2000-OptiMEM mixture was combined at a 1:2 volume ratio with the prepared siRNA-OptiMEM mixture. The volume of the siRNA-LF2000 complex solution corresponded to one-third of the total volume per transfected well. After incubation for 15 min at rt, the siRNA-LF2000 complex solution was added dropwise to the cell culture media. The transfected cells were incubated for 6 h (37 °C, 5% CO_2_) and washed with 1× PBS, before the addition of complete RPMI-1640 medium for cell recovery overnight. The polarization of the transfected M-MΦ was performed on day 6, as described above.

### 2.6. Relative Gene Expression Analysis Using Quantitative Real-Time PCR

The isolation of total RNA from MΦ lysates was performed in accordance with previous procedures [24]. The concentration and quality of the utilized RNA solutions were analyzed using the NanoDrop 2000 spectrophotometer (Peqlab, Erlangen, Germany), considering the ratio of absorbance at 260 nm and 280 nm (A260/A280) or 230 nm (A260/230) of the RNA sample. Only those RNA samples with A260/A280 and A260/230 ratios of 1.90–2.2 were used for the subsequent reverse transcription of 500 ng RNA to cDNA.

The first-strand cDNA synthesis was performed using the SuperScript II Reverse Transcriptase kit (Thermo Fisher Scientific, Waltham, MA, USA) and oligo(dT)_12-18_ primers (Biomers, Ulm, Germany). The obtained cDNA solution (38 μL) was diluted to a 1:5 volume ratio with PCR-grade water (Roche, Basel, Switzerland), prior to carrying out the quantitative real-time (qRT)-PCR analysis. SYBR Green I dye-based reaction mixture preparation and instrument settings were adjusted, as detailed earlier [24].

The nucleotide sequences used for the gene expression analysis are listed in Appendix A. The relative gene expression levels of the target gene transcripts were calculated according to the mathematical model that was established by Pfaffl and colleagues [30], including a reference gene index that consisted of the three reference genes *SDHA*, *RPL13A* and *B2M*, for data normalization, and for considering the qRT-PCR amplification efficiency of the genes.

### 2.7. Cell Characterization with Flow Cytometry

The phenotypic cell characterization was performed with flow cytometry. For cell collection, the MΦ monolayer was washed with 1× PBS and incubated with 1× PBS on ice for 3 h. The loosened cells were rinsed from the culture dish via pipetting, and transferred into tubes. The cell culture dish was rinsed with 1× PBS, in order to collect residual cells. The collected cell suspension was centrifuged at 200× *g* for 10 min, and the cell pellet that was obtained was resuspended in FACS buffer consisting of 5% (*v*/*v*) FBS and 2 mM EDTA (Merck, Darmstadt, Germany) in 1× PBS.

An amount of 0.25∙10^6^ cells were blocked with 100 μg/mL Fc block (BD Bioscience, Franklin Lakes, NJ, USA) and supplemented FACS buffer for 25 min. The cells were then incubated with the antibodies CD14-FITC, CD80-PE-Cy7, CD163-PerCP-Cy5.5, CD206-APC, HLA-DR-APC-H7 (BD Bioscience, Franklin Lakes, NJ, USA) and CD16-PE (BioLegend, San Diego, CA, USA) for 30 min at 4 °C, using the manufacturer’s recommended antibody dilutions. After washing with FACS buffer, flow cytometry was performed with a FACS CANTO II flow cytometer (BD Bioscience, Franklin Lakes, NJ, USA). Regarding the analysis of monocytes from PBMC isolation or enriched cells, 0.5∙10^6^ cells were analyzed using the following antibodies: CD14-FITC, HLA-DR-APC-H7, CD45-PerCP (BD Bioscience, Franklin Lakes, NJ, USA) and CD16-PE (BioLegend, San Diego, CA, USA). The remaining staining procedure did not differ from staining macrophages, as described above. Data were analyzed with Kaluza Analysis Software (Version 2.1; Beckman Coulter, Brea, CA, USA). The arithmetic mean fluorescence intensity (MFI) of each marker was measured, and the autofluorescence of unstained equivalently polarized macrophages was subtracted (∆MFI). The compensation matrix to correct the spectral overlap was calculated from single-stained controls. The isotype and fluorescence-minus-one controls were included, in order to verify the specificity of the staining.

### 2.8. Cytokine Detection Using Immunoassay Analysis

The determination of IL-6 concentrations in cell culture supernatants of polarized MΦ was performed automatically with the Cobas E411 Analyzer (Roche, Basel, Switzerland). Sample values were normalized to the respective sample’s DNA content. The isolation and quantification of the DNA were performed as detailed previously [24].

### 2.9. Immunofluorescence Microscopy

The polarized primary MΦ were fixed with a 2.5% (*w*/*v*) paraformaldehyde solution that was diluted in 1× PBS at 37 °C for 20 min, prior to antibody staining for immunofluorescence analysis of the α-SMA expression. The cell monolayer was washed and stored in 1× PBS at 4 °C until staining. The staining procedure was performed at room temperature. Three washing steps using 1× PBS were performed after each step, as described below. The negative control samples containing only the secondary antibodies were included for every polarized macrophage subtype.

The cell samples were permeabilized for 10 min using 0.1% (*v*/*v*) Triton X-100 in 1× PBS. Thereafter, samples were blocked with 5% (*v*/*v*) normal goat serum (Sigma-Aldrich, St. Louis, MO, USA) in 1× PBS for 1 h, in order to prevent nonspecific antibody binding. Sample incubation with the primary antibodies (monoclonal mouse anti-human α-SMA antibody clone 1A4 [1:50; Agilent, Santa Clara, CA, USA], polyclonal rabbit anti-β-tubulin antibody [1:400; abcam, Cambridge, UK]) was performed, with dilution in blocking solution for 2 h under constant agitation. Next, the samples were incubated with fluorochrome-conjugated secondary antibody (1:200; goat anti-mouse IgG [Alexa Fluor 488; abcam, Cambridge, UK]; 1:800, goat anti-rabbit IgG [Alexa Fluor 555, abcam, Cambridge, UK]) solutions prepared in blocking solution for 1 h in the dark. Counterstaining was performed using 4′,6-diamidino-2-phenylindole (DAPI, 1:200 in 1× PBS; Thermo Fisher Scientific) for 10 min in the dark. After rinsing the slice with 1× PBS, the coverslip was mounted with Roti-Mount FluorCare (Carl Roth, Karlsruhe, Germany). Immunofluorescence images were captured with a Keyence fluorescence microscope (BZ-X810, Osaka, Japan), and the corrected total cell fluorescence of α-SMA was calculated using ImageJ software (Version 1.52a).

### 2.10. Mass Spectrometric XT-I Activity Assay

The relative quantification of the extracellular XT-I activity from cell culture supernatants and the intracellular XT-I activity from cell lysates was performed using a mass spectrometric enzyme activity assay, following a previously published procedure [31,32]. This relative in silico XT-I activity measurement relied on XT-I-catalyzed xylose incorporation into an XT-I-selective acceptor peptide after a defined period.

Cell lysates for intracellular XT-I quantification were obtained by incubating the macrophage monolayer with Nonidet P-40 Substitute (Roche, Basel, Switzerland) containing lysis buffer (250 μL), followed by additional incubation and sample preparation steps, as described previously [24]. Additional cell-free cultivation vessels with complete RPMI-1640 media (2 mL) were prepared and handled in accordance with cell-containing approaches, in order to determine the extracellular XT-I activity from the cell culture supernatant that is based solely on the macrophage-mediated secretion of XT-I to the extracellular space. The XT-I activity that was determined from the cell-free setups was as a result of the FBS-supplementation of the growth medium was used for background subtraction from the extracellular activity values of the vessels that contained macrophages. A prediluted xylosylated-peptide standard of known concentration was prepared for the mass spectrometric determination of XT-I sample activity, as previously described [31], and was included in every mass spectrometric run that was performed. An aliquot of a positive control sample, consisting of recombinant Chinese hamster ovary-derived XT-I protein, was included in every mass assay for inter-assay correction purposes, such as varying reaction mixture compositions. The determined cellular XT-I activity was expressed in enzyme units U (μmol/min), and was normalized to the total protein content of the respective lysate, that was quantified via a bicinchonic acid assay, as specified previously [31].

### 2.11. Data Analysis

The statistical analyses were performed using GraphPad Prism 9 (Version 9.0.0). The Shapiro–Wilk normality test was performed for normal distribution testing. The data were analyzed using a one-way analysis of variance (ANOVA), followed by Tukey’s multiple comparison tests, and expressed as mean ± standard error (SEM). All of the experiments were carried out with the level of replications indicated: the number of independent donor-derived primary cell cultures (n, sample), the number of cell culture replicates created per donor (n, biological), and the number of measurements performed per cell culture replicate (n, technical).

## 3. Results

### 3.1. Characterization of the Monocyte-Derived Macrophages Generated

Previous studies highlighted the importance of macrophage polarization during tissue repair and fibrosis development. We explored whether the expression of the former postulated myofibroblast marker XT-I is affected by macrophage polarizations, on the basis of a simplified in vitro model for the generation of primary human MΦ.

Untouched monocytes were isolated from PBMC via negative selection for MΦ generation. The monocyte content of the sample was assessed using an FCS/SSC plot of an automated hematology analyzer. Any impurities from other cell types of up to 20% were neglected, as these were removed during cell cultivation. We adopted the gating strategy of Marimuthu and colleagues [7], in order to characterize the monocyte subset within the monocyte sample via flow cytometry. The monocyte subsets of the representative sample shown were 90.2% classicals (CD14^++^CD16^−^), 2.93% intermediates (CD14^++^CD16^+^) and 6.52% nonclassicals (CD14^+^CD16^++^), and resembled the monocyte subsets that were found in the corresponding PBMC sample before magnetic cell separation (Appendix A). Considering that both M-CSF and GM-CSF induce MΦ differentiation with different gene expression signatures [13,33], we decided to generate GM- and M-MΦ for comparison, but to concentrate on M-MΦ for further analysis. A representative image of one donor-derived primary M0 M-MΦ culture at different time points is shown in Appendix A.

Stimulation with IFN-γ and LPS or IL-4 was performed for 48 h for the subsequent macrophage polarization towards an M1- and M2-like phenotype, respectively. Since GM-CSF was shown to induce macrophage differentiation towards an M1-like phenotype or dendritic cell differentiation [13,34], those cells were not polarized towards M2-like macrophages using IL-4. As a preliminary step to determine the *XYLT1* mRNA expression and XT-I activity in polarized human macrophages, the functionality and phenotype of the MΦ generated were verified by measuring inflammatory IL-6 secretion and flow cytometric analysis of the surface markers HLA-DR, CD14, CD16, CD163, CD80 and CD206, using the gating strategy shown in Appendix A. We summarized the expression levels expected from our literature-based analysis in Table A1, in order to improve understanding of the results presented below.

HLA-DR is a surface protein that is constitutively expressed by antigen-presenting cells that can be induced by IFN-γ [35,36]. Accordingly, it was used in our study to distinguish M1-like MΦ from unpolarized MΦ. All macrophage subtypes generated express comparable levels of surface marker HLA-DR, except for IFN-γ/LPS-polarized (M1) MΦ, which show the highest MFI of HLA-DR, irrespective of M-CSF- or GM-CSF maturation. The M1 M-MΦ possessed a higher HLA-DR MFI compared to M0 M-MΦ (1.8-fold, *p* = 0.02), while M1 GM-MΦ showed increased HLA-DR expression compared to M0 GM-MΦ (1.6-fold, *p* = 0.02) (Figure A1A).

The analysis of the surface markers CD14 and CD16 showed significantly decreased expressions of both markers in M1 M-MΦ, in comparison with M0 M-MΦ (CD14, 68.6%, *p* < 0.0001; CD16, 90.4%, *p* < 0.0001). M2 M-MΦ also showed a significantly lower MFI of CD14 and CD16 compared to that of M0 M-MΦ (CD14, 46.6%, *p* < 0.0001; CD16, 38.1%, *p* = 0.04); however, it showed higher CD14 and CD16 expressions compared to those of M1 M-MΦ (CD14, 1.7-fold, *p* = 0.002; CD16, 6.4-fold, *p* = 0.006). In addition, the CD14 and CD16 expressions of M1 GM-MΦ and M0 GM-MΦ were not different from each other (Figure A1B,C).

Hemoglobin scavenger receptor CD163 is a macrophage-specific protein, and is a reliable marker for the in vitro discrimination of classically activated macrophages from M0 and M2 MΦ, since it is suppressed by pro-inflammatory stimuli including IFN-γ and LPS [16,37,38]. Furthermore, it can be used in combination with other markers to distinguish between M2 subgroups [11]. Compared to M0 M-MΦ, M1 M-MΦ showed a significantly decreased CD163 surface expression (88.8%, *p* = 0.02); meanwhile, the MFI of CD163 on M2 M-MΦ did not differ from that on M0 M-MΦ. In addition, no differences were found by comparing the CD163 expression of M2 M-MΦ with that of M1 M-MΦ. The MFI of CD163 on M0 GM-MΦ did not differ significantly from that on M1 GM-MΦ (Figure A2A).

Regarding CD80, a positive marker for the in vitro generation of classical macrophages [39], a significantly increased MFI was observed in M1 M-MΦ compared to that in M0 M-MΦ (8.1-fold, *p* < 0.0001). The CD80 MFI of M2 M-MΦ did not differ from M0 M-MΦ, but was significantly decreased compared to that of M1 M-MΦ (78%, *p* = 0.0002) (Figure A2B).

The positive marker for alternative activated macrophages CD206 [39] did not differ significantly between M1 M-MΦ and M0 M-MΦ (*p* = 0.5), while M2 M-MΦ possessed a significantly higher CD206 MFI than that for M0 M-MΦ (2.6-fold, *p* = 0.001). The CD206 surface expression of M2 M-MΦ was also significantly higher compared to that of M1 M-MΦ (5.0-fold, *p* = 0.0001). In addition, a significantly decreased CD206 surface expression was found in M1 GM-MΦ relative to M0 GM-MΦ (48.1%, *p* = 0.02) (Figure A2C).

Classically activated MΦ possess a pro-inflammatory phenotype that is characterized by the expression of inflammatory cytokines, co-stimulatory molecules, such as CD80, and MHC II molecules, including HLA-DR, which promotes and amplifies the T_H_1 polarization of T lymphocytes [11]. The concentration of the MΦ-derived prototypical inflammatory cytokine IL-6 in the cell culture supernatants was measured for their functional characterization of M1-like MΦ, and to distinguish them from the other MΦ subtypes. All of the macrophage subtypes generated expressed comparable levels of IL-6 production, except for M1 M-MΦ, which showed a significantly higher IL-6 secretion compared to that for M0 M-MΦ (103.5-fold, *p* = 0.03) or M2 M-MΦ (100.3-fold, *p* = 0.03). By contrast, M1 GM-MΦ showed a higher but not statistically significant IL-6 production, in comparison to that of M0 GM-MΦ (59.1-fold, *p* = 0.1) (Figure A2D).

A complete overview of the phenotype characterization of generated MΦ subtypes is shown in Table 1.

It can be concluded from the initial results that the phenotype of the M-MΦ subtypes in our model system resembled those of classical and alternative activated macrophages. The activation state markers that were analyzed could be used for the discrimination of M1 M-MΦ from M0 M-MΦ or/and M2 M-MΦ, or to identify M2 M-MΦ from M0 or M1 M-MΦ. Furthermore, the relative changes in the surface expression of the markers analyzed or in IL-6 secretion were visible in both M1 M- and GM-MΦ, relative to the M0 MΦ, but were significantly more pronounced in the M-MΦ.

It has been suggested that the assessment of MΦ polarization cannot rely solely on a few cell surface markers, but should be addressed via gene expression analysis [40]. Thus, we continued investigations with a genotypic characterization of the MΦ subtypes generated. The expressions of the pro-inflammatory genes *IL1B* and *IL8* were chosen for the identification of classically activated MΦ, and the expression of the ECM components *HSPG2* and *SDC2* were chosen for the discrimination of alternative activated MΦ (Figure A3). The expression levels expected from our literature-based analysis of these genotypic MΦ polarization state markers can be found in Table A1.

The M1 M-MΦ generated in our model system showed significant 3.1-fold (*p* = 0.01) and 17.3-fold (*p* = 0.003) increases in the mRNA expressions of the inflammatory cytokines *IL1B* and *IL8*, respectively, compared to M0 M-MΦ; in contrast, the *IL1B* and *IL8* expressions of M2 M-MΦ were reduced by 91.1%, (*p* = 0.6) and 52.3% (*p* > 0.9999), respectively, relative to those of M0 M-MΦ. In comparison to M1 M-MΦ, the expression levels of *IL1B* (*p* = 0.0001) and *IL8* (*p* = 0.002) were each suppressed by 97.2% in M2 M-MΦ. Similarly to M-MΦ, M1 GM-MΦ showed 15-fold (*p* < 0.0001) and 8-fold (*p* < 0.0001) increases in gene expressions of *IL1B* and *IL8*, respectively, compared to those of M0 GM-MΦ (Figure A3A,B).

High sulfated HSPG produced by macrophages themselves can modify their phenotypes by effecting the bioavailability of IFN-γ [10]. Interestingly, all the MΦ generated only expressed marginal mRNA levels of the CSPGs *VCAN* and *ACAN* (data not shown). We could show that M1 M-MΦ decreased the expression of PG *HSPG2*, that has been associated with an M2-like phenotype [18], by 89.8% (*p* = 0.3), and that of *SCD2* by 90.4% (*p* = 0.003) compared to M0 M-MΦ. By contrast, M2 M-MΦ showed a 3.7-fold (*p* < 0.0001) increase in induced *HSPG2* and a 1.5-fold (*p* = 0.2) higher *SDC2* expression compared to M0 M-MΦ. Consequently, the *HSPG2* and *SDC2* expressions of M2 M-MΦ were higher by 36.2-fold (*p* < 0.0001) and 15.9-fold (*p* < 0.0001), respectively, compared to that of M1 M-MΦ. Compared to M0 GM-MΦ, the M1 GM-MΦ showed a significantly reduced *HSPG2* and *SDC2* expression by 88% (*p* < 0.0001) and 68.7% (*p* = 0.0002), respectively (Figure A3C,D). The results of this genotypic characterization are shown in Table 2.

In summary, the results of the phenotypic, functional (Figure A1, Figure A2, Table 1) and genotypic (Figure A3, Table 2) MΦ subtype characterizations showed a successful in vitro generation of primary M1- and M2-like MΦ cultures that can be used for the downstream analysis of XT-I expression regulation.

### 3.2. XT-I Reveals a Marker for M1- and M2-Macrophages

Having shown that the primary cell model gives rise to M1- and M2-like MΦ from PBMC with distinct phenotypic and genotypic characteristics, we elucidated the expression regulation of the former postulated myofibroblast marker XT-I in human primary MΦ, on the basis of polarization by qRT-PCR analyses, and on an XT-I activity assay (Figure 1).

The *XYLT1* mRNA expression of M1 M-MΦ was significantly reduced by 71.4% (*p* = 0.008), while that of M2 M-MΦ significantly increased 2.1-fold (*p* < 0.0001), in comparison to that of M0 M-MΦ. In comparison to M1 M-MΦ, M2 M-MΦ significantly upregulated 7.2-fold (*p* < 0.0001) relative *XYLT1* mRNA expression. Consistently with M1 M-MΦ, the *XYLT1* expression of M1 GM-MΦ significantly reduced by 89.4% (*p* = 0.0001) relative to M0 GM-MΦ (Figure 1A). In contrast to the differential expression of the *XYLT1* isoform in polarized MΦ, the expression of *XYLT2* did not differ significantly among the different MΦ subtypes (Figure 1B). It is noteworthy that, compared to the IL-4-mediated *XYLT1* mRNA induction in M-MΦ, the IL-4 stimulation of primary normal human fibroblasts with a comparable cytokine concentration of 20 ng/mL and polarization time of 48 h decreased the relative *XYLT1* mRNA expression, instead of inducing it (data not shown).

The relative change in the *XYLT1* mRNA expression of M1 M-MΦ compared to M0 M-MΦ resembles that of M1 GM-MΦ compared to M0 GM-MΦ (Figure 1A); therefore, we decided to solely determine the cellular XT-I activity of M-MΦ subtypes after a polarization time of 48 and 72 h, and neglect that of the GM-MΦ subtypes. The M1 M-MΦ showed reduced extracellular XT-I activity (55.9%, *p* < 0.0001) after a polarization time of 48 h, while M2 M-MΦ possessed a 1.8-fold upregulated XT-I activity (*p* < 0.0001) compared to M0 M-MΦ. Therefore, the extracellular XT-I activity of M2 M-MΦ was 4.2 times (*p* < 0.0001) higher than that of M1 M-MΦ after a polarization period of 48 h. The corresponding intracellular XT-I activities of the M-MΦ subtypes were significantly lower (*p* < 0.0001) compared to their extracellular activities. As such, the intracellular XT-I activity of M0 M-MΦ reduced by 72.6% (*p* < 0.0001), compared to its extracellular activity. Regarding the intracellular activity of M1 M-MΦ, a 74.9% decrease (*p* = 0.1) in enzyme activity was detectable, whereas the intracellular XT-I activity of M2 M-MΦ was 1.7 times higher (*p* = 0.2) compared to M0 M-MΦ after polarization for 48 h. In addition, the intracellular activity of M2 M-MΦ was 6.6 times higher (*p* < 0.0001) than that of M1 M-MΦ (Figure 1C). After a polarization period of 72 h, we observed comparable results to those that were obtained after 48 h of polarization. The extracellular XT-I activity of M1 M-MΦ significantly reduced by 37.4% (*p* < 0.0001), and that of M2 M-MΦ significantly increased by 1.5 times (*p* < 0.0001), in comparison to M0 M-MΦ. Accordingly, M2 M-MΦ exhibited a 2.3 times higher (*p* < 0.0001) extracellular XT-I activity compared to M1 M-MΦ. Consistently with the cellular XT-I activity of M0 M-MΦ that was determined 48 h post-polarization, the intracellular XT-I activity of M0 M-MΦ 72 h post-polarization significantly decreased by 79.6% (*p* < 0.0001) compared to the corresponding extracellular enzyme activity. The intracellular activity of M1 M-MΦ 72 h post-polarization reduced by 66.7% (*p* = 0.07), while the intracellular activity of M2 M-MΦ was not significantly changed (1.3 times, *p* = 0.9), in comparison to that of M0 M-MΦ. When compared directly to that of M1 M-MΦ, the intracellular XT-I activity of M2 M-MΦ was 3.8 times higher (*p* = 0.002) after a polarization period of 72 h (Figure 1D).

Together, these results clearly show that XT-I is regulated differently by macrophage polarization, leading to distinct *XYLT1* and cellular XT-I activity differences in M1- and M2-like M-MΦ, while the *XYLT2* isoform expression in MΦ is independent of the polarization status. Thus, the expression of the *XYLT1* isoform and cellular XT-I activity reveal a suitable genotypic and enzymatic marker for the differentiation of in vitro generated MΦ subtypes for studying tissue macrophages.

### 3.3. XT-I Expression Is Regulated by TGF-β1 Signalling Components in Primary MΦ

We determined the expressions of *TGFB1* and *SMAD7*, which are known XT-I regulatory factors [24], in order to further analyze the underlying regulatory mechanism for *XYLT1* and XT-I activity differences of M1- and M2-like MΦ (Figure 2).

The expression of *TGFB1* was comparable between M0 M-MΦ and M2 M-MΦ; it was significantly reduced in M1 M-MΦ by 67.4% (*p* < 0.0001) and in M1 GM-MΦ by 52.3% (*p* < 0.0001), compared to the respective M0 GM- or M-MΦ. Comparing M2 and M1 M-MΦ showed a 3.0-fold higher (*p* < 0.0001) *TGFB1* expression in M2 M-MΦ (Figure 2A). Thus, the relative changes in the basal *TGFB1* expressions of M0-, M1- and M2-like M-MΦ may have contributed to the differences in *XYLT1* expression and cellular XT-I activity observed. Smad7 is an intracellular inhibitor of the TGF-β signaling pathway that regulates inflammatory cytokine responses. Thus, a high Smad7 expression was associated with defective TGF-β1 signaling and inflammatory cytokine production [41]. The relative gene expression of *SMAD7* was 1.2 times higher in M1 M-MΦ, and significantly suppressed by 37.7% (*p* = 0.0001) in M2 M-MΦ compared to M0 M-MΦ. In direct comparison to M1 M-MΦ, the *SMAD7* expression of M2 ΦMΦ was significantly reduced by 47.1% (*p* < 0.0001). In addition, the relative *SMAD7* expression of M1 GM-MΦ was 1.4 times higher (*p* = 0.0001) compared to that of M0 GM-MΦ (Figure 2B).

To conclude, the decreased *XYLT1* mRNA expression and XT-I activity of M1-like MΦ could be caused by a decreased basal *TGFB1* expression, and by a simultaneous *SMAD7* induction. There were no significant *TGFB1* expression differences between M0- and M2 M-MΦ; thus, the relative increase in XT-I expression observed in M2-like MΦ could be due to the lower inhibitory *SMAD7* expression seen in these cells.

### 3.4. XT-I Expression and Activity Resembles an Early Marker for Assessing MMT

The protein or mRNA expression of *ACTA2* is generally used to identify myofibroblasts in tissues, or in the context of MMT [42,43,44]. Since XT-I is a formerly postulated marker for the fibroblast-to-myofibroblast transition, and identified here as a genetic and enzymatic positive marker for in vitro generated M2-like MΦ, we wanted to determine whether the M2-like MΦ that was generated showed further myofibroblast characteristics. We, therefore, analyzed the mRNA expression of *ACTA2*, and α-SMA protein expression as an additional myofibroblast marker in M-MΦ subtypes, using qRT-PCR and immunofluorescence analyses (Figure 3).

Our results showed that polarized M-MΦ subtypes express the myofibroblast marker *ACTA2* with a slightly but not significantly higher relative expression level in M1 M-MΦ, by 1.7-fold (*p* = 0.8), and M2 M-MΦ by 2.1-fold (*p* = 0.3), compared to M0 M-MΦ. Accordingly, the relative *ACTA2* expression of M2 M-MΦ was 1.3 times higher (*p* = 0.9) compared to that of M1 M-MΦ. In contrast to M1 M-MΦ, the expression of *ACTA2* decreased by 24.7% (*p* = 0.6) in M1 GM-MΦ compared to M0 GM-MΦ, but did not reach statistical significance (Figure 3A). Representative images of the immunofluorescence staining of α-SMA in M-MΦ subtypes are shown in Figure 3C. The results of the immunofluorescence analysis were consistent with the gene expression analysis, showing no distinct differences in the α-SMA protein level of the M0 and M1-MΦ subtypes that were generated (1.2-fold; *p* = 0.5), but a 2.1-fold increase (*p* < 0.0001) in corrected total cell fluorescence of α-SMA in M2 M-MΦ compared to M0 M-MΦ. Compared to M1 M-MΦ, the α-SMA expression of M2 M-MΦ was 1.7 times higher (*p* = 0.0004) (Figure 3B). It can be summarized that all M-MΦ subtypes showed *ACTA2* and α-SMA expression. The highest α-SMA expression was observed in M2 M-MΦ, indicating MMT.

### 3.5. Primary Macrophage Polarization Is Independent of XT-I Expression

The current study identified that macrophage polarization leads to changes in the relative *XYLT1* mRNA expression, and in the cellular activity of polarized macrophage species. We established a suitable cell culture model to investigate the impact of diminished XT-I expression and activity on macrophage polarization and function, using siRNA-mediated *XYLT1* knockdown in M-MΦ, with the absence and presence of M1 and M2 stimuli (Figure 4).

The transfection of M0 M-MΦ with an siRNA that targeted *XYLT1* (si*XYLT1*) significantly reduced *XYLT1* mRNA expression by 85.1% (*p* < 0.0001) relative to M0 M-MΦ that was transfected with a non-targeting negative control siRNA (siNC). As M1 M-MΦ already possessed significantly diminished *XYLT1* mRNA expression compared to M0 M-MΦ (Figure 1A), the si*XYLT1* transfection did not statistically significantly reduce the *XYLT1* expression of M1 M-MΦ compared to siNC-transfected M1 M-MΦ; however, a 73.6% reduction (*p* = 0.5) was observed. In contrast, the siRNA-mediated *XYLT1* knockdown in M2 M-MΦ reduced the relative *XYLT1* expression by 80.6% (*p* < 0.0001) compared to that of siNC-transfected cells (Figure 4A). The specificity of the applied *XYLT1* knockdown was confirmed by measuring the *XYLT2* isoform expression of si*XYLT1*-transfected M0, M1 or M2 M-MΦ and siNC-transfected cells, which did not differ from each other statistically (Figure 4B). The siRNA-mediated *XYLT1* knockdown in M0 M-MΦ significantly reduced extracellular XT-I activity by 80.8% (*p* < 0.0001) compared to that in siNC-transfected M0 M-MΦ (Figure 4C,D). The extracellular XT-I activity of si*XYLT1*-transfected M1 M-MΦ significantly decreased by 74.9% (*p* < 0.0001) compared to that of siNC-transfected M1M-MΦ. Regarding the intracellular XT-I activity of si*XYLT1*- and siNC-transfected M0 M-MΦ, diminished activity was observed (by 72.5%, *p* < 0.0001) in si*XYLT1*-transfected M0 M-MΦ (Figure 4C,D), while the intracellular XT-I activity of *XYLT1*-silenced M1 M-MΦ did not differ statistically from that of siNC-transfected M1 M-MΦ, which showed a reduction of 77.8% (*p* = 0.4) (Figure 4C). The siRNA-mediated *XYLT1* knockdown in M2 M-MΦ significantly reduced extracellular XT-I activity by 80.6% (*p* < 0.0001) and intracellular XT-I activity by 72.5% (*p* < 0.0001) compared to siNC-transfected M2 M-MΦ (Figure 4D). These results confirmed that siRNA-mediated *XYLT1* silencing of activated M-MΦ was successfully applied, leading to significantly diminished XT-I expression and cellular activity levels in the cell culture model.

In order to evaluate the impact of diminished XT-I expression on macrophage polarization, the established *XYLT1* knockdown model was used to characterize the polarization state of si*XYLT1*-transfected M-MΦ relative to siNC-transfected cells. Thus, the selection of MΦ polarization state markers (Table A1) was analyzed through qRT-PCR analysis (Figure 5).

We decided to shorten the polarization time after siRNA-mediated *XYLT1* knockdown to 24 h, instead of the 48 h previously used to observe the primary transcriptional effects and minimize the secondary effects of autocrine mediators on the macrophage polarization. The analysis of the two representative M1-like marker genes *IL1B, IL8* (Figure 5A,B) and M2-like markers *HSPG2* and *CD206* (Figure 5C,D) in *XYLT1*-silenced M0, M1 or M2 M-MΦ did not show significant expression differences compared to the respective siNC-transfected controls. Interestingly, the cytokine *IL1B* expression of si*XYLT1*-transfected M0 M-MΦ was 2.7 times higher (*p* = 0.9998) compared to siNC-transfected M0 M-MΦ. In addition, the marker expressions of *SDC2*, *ACTA2* and *TGFB1* (Appendix A–C), as well as the secretion of pro-inflammatory cytokine IL-6 (Appendix A) in *XYLT1*-silenced M0, M1 or M2 M-MΦ did not differ significantly from that of the respective siNC-transfected controls. Consistently with the data mentioned above, we observed an *SDC2* expression that decreased by 45.3% (*p* = 0.05), a 1.4-fold increase in *ACTA2* and an IL-6 expression that decreased by 28% (*p* > 0.9999) after the siRNA-mediated *XYLT1* knockdown of M0 M-MΦ (Appendix A).

The present data suggest that macrophage polarization is independent of XT-I isoform expression and activity, whereas a diminished *XYLT1* expression and XT-I activity in M0 M-MΦ seemed to have a slight impact on the basal expressions of some polarization state markers.

## 4. Discussion

Considering the complex mechanisms that are involved in fibroproliferative diseases, macrophages are a crucial cell type which modulate the fibrotic process and its resolution [11]. Increased serum XT activity and/or cellular *XYLT1* expression have been identified as biochemical markers for assessing fibrotic remodeling processes, such as the cytokine-mediated fibroblasts-to-myofibroblasts transition in SSc [21,27,28,45]. Given the missing link between fibrotic XT activity increase and XT regulation in immune cells, this study aimed to investigate the impacts of macrophage polarization on XT-I expression and activity. An overview of the results obtained in this study, and the relationship between PBMC, macrophages, XT-I, PG and GAG is shown in Figure 6.

Since in vitro generated M-MΦ from patients with SSc have been shown to express surface markers that are associated with M1 and M2 polarization [46,47], we decided to use a simplified primary M-MΦ cell model that mimicked the MΦ subtypes found in prototypical inflammatory fibrotic disease SSc. Accordingly, PBMC from healthy blood donors were subjected to a negative selection of CD14^+^ and/or CD16^+^ monocytes. The MΦ generated were polarized in the presence of LPS/IFN-γ or IL-4 toward M1-like and M2-like MΦ subpopulations, respectively. The efficacy of MΦ polarization was confirmed by the analysis of surface markers, IL-6 cytokine secretion and the gene expression of widely accepted MΦ activation state markers. Consistently with the findings from previous studies summarized in Table A1, our study observed higher expressions of M2 phenotype markers CD14, CD16 and CD206 in M2 compared to M1 M-MΦ. Furthermore, our results showed more pronounced M1-like marker HLA-DR, CD80 and IL-6 expressions of M1 M-MΦ compared to those of M0 and M2 M-MΦ cultures, and were also in line with data reported from the literature. In addition, we observed a negative correlation of HLA-DR expression with that of CD163 in our cell culture model, which is in close agreement with the literature of tumor-associated macrophages [48]. It is noteworthy that we did not find CD163 upregulation in M2 M-MΦ compared to the other M-MΦ subtypes, although CD163 was defined as a surface marker for M2-like cells [33]. Our observation that CD163 was not upregulated in M2 M-MΦ is consistent with previous reports that showed M2 M-MΦ mainly increased CD206 expression, but not for CD163 [49]. Previous research has shown that IL-10 stimulation induces CD163 and *TGFB1* expressions, which can be used to distinguish in vitro generated M2a and M2c MΦ subtypes [11,50,51,52,53]; therefore, we concluded that M2 M-MΦ cultures in the present study belong to the M2a MΦ subtype. This hypothesis was further strengthened by preliminary data (not shown) that demonstrated suppressed *XYLT1* expression in IL-10-stimulated M-MΦ, and by the *TGFB1* expression analysis of M2 and M0 M-MΦ cultures in this study, which did not significantly differ from one another.

As a result of the high plasticity of human macrophages, we performed an additional genotypic characterization of GM- and M-MΦ cultures, in order to assess the full spectrum of macrophage polarization in our in vitro model. In agreement with the reported findings from previous studies, we observed an increase in inflammatory cytokine *IL1B* and *IL8* expressions in M1 GM- and M1 M-MΦ compared to those in other MΦ subtypes generated. As macrophage polarization is associated with changes in the gene and protein expressions of HSPGs, resulting in a 2-fold increase inHS-GAG content in M2-like MΦ cultures compared to M1-like MΦ [54], we analyzed the expressions of representative HS core proteins *HSPG2* and *SDC2* in our in vitro model. In contrast to earlier studies that were summarized by Swart and Troeberg, which reported no regulation of ECM PG *HSPG2* expression by in vitro macrophage polarization [14], we revealed a decrease in *HSPG2* expression upon M1 polarization, and an increased expression level upon M2 stimulation of M-MΦ. These data are partly consistent with those obtained from Witherel and colleagues, who demonstrated the highest *HSPG2* expression in human M2-like MΦ [53]. According to microarray data, *SDC2* expression was higher in diseases with an M2-like phenotype, such as asthma, while in vitro generated M1-like macrophages possessed downregulated *SDC2* expression [14]. Our cell culture model similarly showed a higher expression of *SDC2* in M2 M-MΦ, while M1 M-MΦ exhibited a lower *SDC2* level compared to the other two M-MΦ subtypes. Taking together all of the data generated from the M-MΦ model, it can be concluded that the in vitro generation of primary M0, M1 and M2 M-MΦ, with distinct phenotypic and genotypic characteristics, was successful. This M-MΦ model was, therefore, used to study the effect of macrophage polarization on XT isoform expression regulation.

The mRNA expression levels of GAG-initiating enzymes *XYLT1* and *XYLT2* have been explored in a large number of human cell lines [55], and studies with primary human cells mainly focused on the regulation of the XT-I isoform in non-immune cells, such as primary fibroblasts, chondrocytes and nucleus pulposus cells [22,56,57]. To the best of our knowledge, only a few studies have briefly depicted XT isoform expression in human immune cells, such as macrophages using microarray technologies, as reviewed by Swart and Troeberg [14,58]. Thus, little is known about XT expression and activity in human primary M1- and M2-like MΦ subpopulations. We found here, for the first time, using our in vitro MΦ culture model, that the *XYLT1* mRNA expression and XT-I activity in MΦ are regulated differently by macrophage polarization. *XYLT1* expression was diminished in M1-like MΦ, while in that of the M2 M-MΦ culture, it was significantly upregulated compared to unpolarized cells. Activation of the TGF-β pathway plays a role in both macrophage polarization and XT-I regulation; therefore, we measured the expression of XT-I suppressor *SMAD7,* which in this study, and in agreement with previous studies [24], showed a negative expression correlation to the expression of *XYLT1*.

Consistently with the results of stimulated human fibroblasts [22,24], we did not detect a regulation of the *XYLT2* isoform by macrophage polarization. We assumed, analogously, that this observation was based on differences in the transcriptional regulation of the *XYLT1* and *XYLT2* promoter regions [59,60]. The missing *XYLT2* expression differences in the M-MΦ subtypes of this study differed from a previous microarray data analysis conducted by Swart and Troeberg, who demonstrated *XYLT2* downregulation or upregulation in IFN-γ- or LPS/IFN-γ-stimulated human macrophages, respectively [14]. We presume that these differences may be due to the differences in the experimental setup chosen, for example, in the type of stimuli and concentration, polarization time, or in the monocyte isolation and detachment procedure performed, that have been discussed elsewhere [49,52,61].

Changes in the relative *XYLT1* expression of M-MΦ subtypes resulted in distinct extracellular XT-I activity changes that were determined 48 and 72 h post-polarization. Similarly to the results from cytokine-stimulated non-immune cells [22,24,27], *XYLT1* expression differences in M-MΦ subtypes led to comparable time-dependent changes in the extracellular MΦ-derived XT-I activity; meanwhile, the intracellular XT-I in MΦ subtypes remained relatively constant over time. In conclusion, we provide first evidence that human macrophages, as representative immune cells, do not only show *XYLT1* expression, but also secrete functional XT-I protein into the extracellular space; therefore, they contribute to the measurable XT activity in human serum or liquid samples [27,28,62]. In view of the still limited number of protein markers that clearly distinguish human M1- and M2-like MΦ, the differential XT-I expression and activity pattern in M1- and M2-like macrophages provide a strong argument for its usage as an in vitro MΦ activation state marker that is suitable for polarization periods of 24 h to 72 h.

The observed decrease in *XYLT1* expression in M1-like MΦ is consistent with published microarray data that show diminished *XYLT1* expression in human M1-like macrophages [14]. Since LPS stimulation of human fibroblasts led to the same results of decreased *XYLT1* expression as that observed here in M1-like MΦ [22], we suggest that the LPS-mediated *XYLT1* decrease may be independent of the cell type. Nonetheless, further analyses are necessary, in order to prove the cell type independence of decreased LPS-mediated *XYLT1* expression. As discussed in the aforementioned study [22], we also hypothesize that the LPS-mediated decrease in *XYLT1* expression in M1-like MΦ is due to the induction of *SMAD7*, which was identified as an XT-I suppressor in gene knockdown experiments [24]. This assumption is further strengthened by the results of the present study, which showed a negative expression correlation between *XYLT1* and *SMAD7* mRNA expressions in M1- and M2-like MΦ cultures. The *XYLT1* expression and activity increase observed in primary M2 M-MΦ cultures in this study is consistent with previous research, which postulated that XT-I expression and activity is a reliable marker for fibroblast-to-myofibroblast differentiation that occurs during fibrotic tissue remodeling [29]. IL-4 stimulation of primary human fibroblasts decreases relative *XYLT1* mRNA expression rather than induces it; therefore, we conclude that cell type-specific differences in IL-4-mediated XT-I expression regulation exist. Nevertheless, the M2 M-MΦ cultures that were generated with induced XT-I expression exhibited a myofibroblast-like phenotype, which was similar to our previous fibroblast-to-myofibroblast transition model [29], as evidenced by the detection of *ACTA2* and α-SMA protein expressions in our study.

To the best of our knowledge, this is the first study to demonstrate a myofibroblast-like phenotype in in vitro generated human primary M2 M-MΦ cultures. The process of MMT was studied extensively in murine models, while studies using human tissue samples have been limited, and those using primary human cells were lacking [5]. As no other comparable study could be found that analyzed α-SMA expression in primary human MΦ subtypes, we compare our results with those of cytokine-stimulated CD14^+^ monocytes from SSc patients and healthy subjects, or with data derived from fibrotic human tissue sections [42,43,63]. Our observation of increased α-SMA expression in in vitro generated human primary M2 M-MΦ cultures is partly comparable to the results from Rudnik and colleagues, which showed increased α-SMA expression in CD14^+^ monocytes from SSc patients or healthy subjects who were stimulated with a profibrotic cytokine mixture consisting of TGF-β, IL-4, IL-10 and IL-13 (10 ng/mL each) [63]. As a result of a lack of cell characterization in the previously mentioned study, it is not known whether the CD14^+^ monocyte cultures differentiated into mature macrophages or not, as M- or GM-CSF were not included in the experimental setup. Furthermore, our results are in accordance with previous studies that utilized the co-immunofluorescence staining of α-SMA and CD206 for the detection of MMT in human renal fibrosis tissue, indicating an M2-like phenotype of α-SMA that expressed myofibroblast-like macrophages [42,43,64]. Therefore, we provide further evidence that human M2-like MΦ may have the potential to transdifferentiate into α-SMA-expressing myofibroblast-like cells that are characterized by upregulated *XYLT1* mRNA expression and XT-I secretion.

Taking the results of this and previous research together, it can be suggested that changes in the expression of linker-initiating enzyme XT-I may have affected the GAG content of M1- and M2-like macrophages reported by Martinez and colleagues [39]. Since reductions in, or alterations of, macrophage-derived HSPG resulted in macrophage activation [10], we addressed this through siRNA-mediated *XYLT1* knockdown experiments to determine whether decreased XT-I expression affected macrophage activation status. Our results show, in agreement with data from previous studies [22,24], that diminished *XYLT1* mRNA levels result in reduced cellular XT-I activities in the cell culture system. Despite detecting slight expression changes in untreated M-MΦ cultures after siRNA-mediated *XYLT1* suppression, which indicated a more M1-like phenotype, we did not detect significant genotypic changes in *XYLT1*-reduced M1- or M2 M-MΦ cultures, compared to controls. Our initial data are in close agreement with the findings in fibroblasts from patients with *XYLT1* mutations, or with those of *XYLT1*^−/−^ fibroblasts generated by CRISPR/Cas9 gene editing that demonstrated decreased CS content or *ACAN* mRNA levels rather than HS content or *SDC2* mRNA levels, respectively [20,65]. Since macrophage polarization has been shown to mainly effect the regulation of HS-associated gene and protein expressions [10,14,17,18], we conclude that the polarization of human MΦ toward an M1- or M2-like phenotype is independent of XT-I expression and activity, considering that XT-I expression is mainly associated with CSPG synthesis. A next step to take would be to fully characterize the generated *XYLT1* knockdown model in terms of the phenotype, including further analysis of surface marker expressions and MMT characteristics. This will allow us to further elucidate the context of altered XT-I expression in immune cells, with potential links to human disorders that have previously been associated with altered XT expression.

Several limitations should be mentioned in the current study. In summary, we used a simplified in vitro M1 (IFN-γ/LPS) and M2 (IL-4) model to investigate the expressions and activities of human XT-I in primary polarized monocyte-derived macrophage subtypes. This M1/M2 model oversimplifies the in vivo macrophage heterogeneity of SSc macrophages, which exhibited both an M1- and M2-like phenotype that was characterized by increased HLA-DR and CD206 surface expressions as well as enhanced IL-6 and TGF-β1 secretions [66]. Furthermore, we did not phenotypically nor genotypically subclassify the M2 M-MΦ subtype in our study into M2a or M2c, even though we assumed the successful generation of an M2a subclass. We only performed a counter test with IL-10-stimulation of M-MΦ promoting the polarization toward the M2c subgroup, showing a different genotype compared to the M2a subclass. Carrying out prospective studies with a larger sample size are essential, in order to further evaluate XT-I expression in different M2 M-MΦ subclasses. As it has been shown that CD14^+^CD16^−^ and CD14^+^CD16^+^ monocytes possessed different characteristics regarding the generation of polarized MΦ [61], the experiments of the present study should be repeated, solely with CD14^+^ monocyte cultures, to evaluate the effects of CD16^+^ monocytes on the characteristics of in vitro generated MΦ. Lastly, we want to mention that the transfection method that utilized Lipofectamine 2000 reagent has been shown to interfere with LPS-mediated cell responses in other systems [67]. We decided to use this transfection reagent, as we could not visualize significant morphological or genotypic changes in our MΦ cultures, with and without transfection. Nonetheless, the use of another transfection method, such as electroporation, should be considered, in order to exclude with certainty that the polycationic lipid-based transfection had no or minimal effects on macrophage polarization. Since we have focused exclusively on the changes in XT-I expression on MΦ polarization, future approaches should address the impact of the XT-II isoform in this context, as a loss of XT-I activity or expression may be compensated for by the other isoform. Additionally, the methods applied can be complemented by histochemical GAG determinations, or HPLC-based analyses, in order to clarify the extent of altered XT expression on MΦ polarization.

## 5. Conclusions

We identified XT-I as a novel macrophage polarization state marker for in vitro generated M1- and M2-like MΦ subtypes. Pro-fibrotic M2-like MΦ cultures exhibited higher XT-I secretion into the extracellular space, as well as increased α-SMA expression, indicating initial MMT. Therefore, XT-I resembles a marker, for not only the fibroblast-to-myofibroblast transition, but also for the MMT that occurs during fibrotic tissue remodeling. It remains unclear why M2-like MΦ express or secrete XT-I, since diminished *XYLT1* expression by siRNA-mediated gene silencing of M2-like MΦ did not affect their pro-fibrotic phenotype. In conclusion, this study provides new insights into macrophage biology and human XT-I regulation in immune cells, and broadens the view of XT-I being a myofibroblast marker in the process of MMT.

## Figures and Tables

**Figure 1 biomedicines-10-02869-f001:**
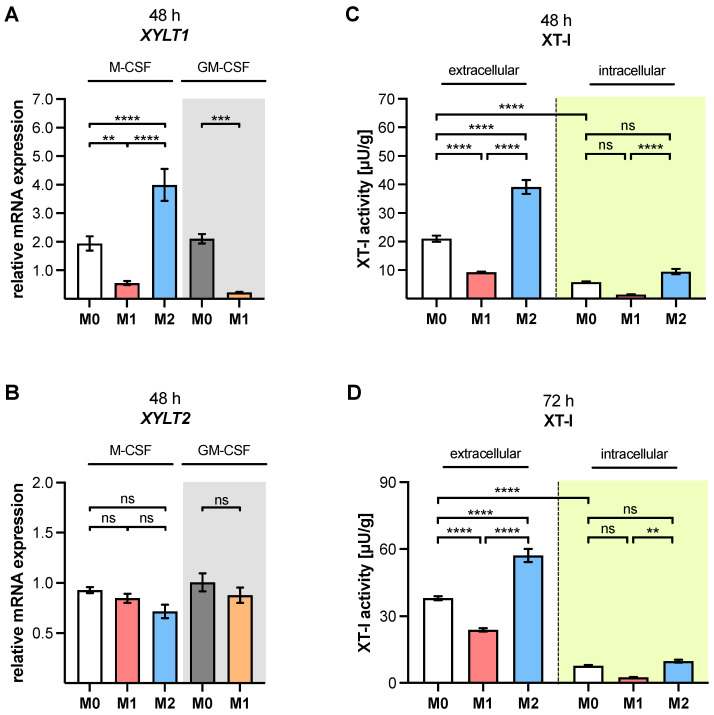
Expressions of *XYLT1* and *XYLT2*, and cellular XT-I activity of polarized macrophages. Negatively selected monocytes were differentiated to macrophages using M-CSF or GM-CSF (highlighted in grey). After 6 days, MΦ were stimulated with IFN-γ/LPS (M1), IL-4 (M2), or with no additive (M0). Cells were harvested after a polarization time of 48 h. The relative gene expressions of (**A**) *XYLT1* and (**B**) *XYLT2* were determined via qRT-PCR analysis (n, sample = 3; n, biological = 2; n, technical = 3). Polarization times of (**C**) 48 h and (**D**) 72 h were applied for cellular XT-I activity determination of polarized M-MΦ (n, sample = 3; n, biological = 3; n, technical = 3) by mass spectrometry. The cell culture supernatant was collected for the extracellular XT-I activity determination, while the cell lysate was used for intracellular XT-I activity (highlighted in yellow) measurement. XT-I activities were referred to the total protein content of the lysates, and are expressed in μU/g. All data were analyzed using one-way ANOVA, followed by Tukey’s multiple comparison tests, and expressed as mean ± SEM. ns (not significant), *p* < 0.01 (**), *p* < 0.001 (***), *p* < 0.0001 (****).

**Figure 2 biomedicines-10-02869-f002:**
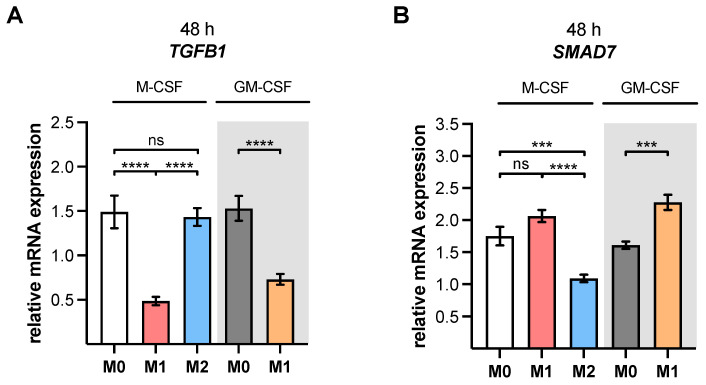
Gene expressions of polarized human monocyte-derived macrophages. Negatively selected monocytes (n, sample = 3; n, biological = 2; n, technical = 3) were differentiated to macrophages using M-CSF or GM-CSF (highlighted in grey). After 6 days, MΦ were stimulated with IFN-γ/LPS (M1), IL-4 (M2), or with no additive (M0). Cells were harvested after a polarization time of 48 h. The relative gene expressions of (**A**) *TGFB1* and (**B**) *SMAD7* were determined using qRT-PCR. The data were analyzed using a one-way ANOVA, followed by Tukey’s multiple comparison tests, and expressed as mean ± SEM. ns (not significant), *p* < 0.001 (***), *p* < 0.0001 (****).

**Figure 3 biomedicines-10-02869-f003:**
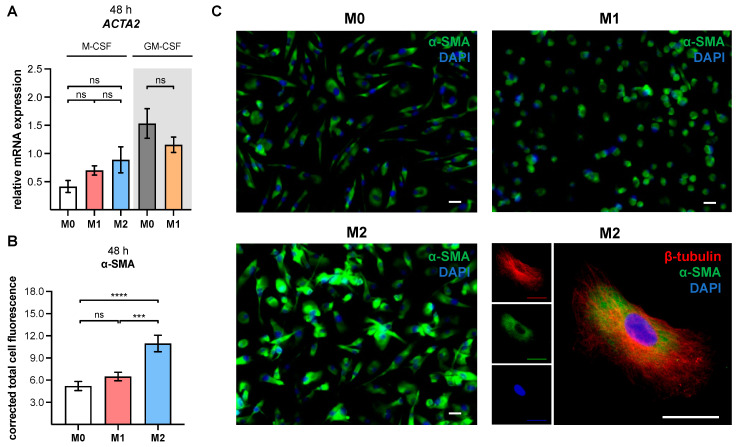
Analysis of an α-SMA gene, and protein expression in polarized human MΦ. Negatively selected monocytes (n, sample = 3; n, biological = 2; n, technical = 3) were differentiated into macrophages, using M-CSF or GM-CSF (highlighted in grey). After 6 days, MΦ were stimulated with IFN-γ/LPS (M1), IL-4 (M2), or with no additive (M0). (**A**) The relative *ACTA2* expressions of the in vitro generated MΦ were determined via qRT-PCR, after a polarization time of 48 h. (**B**) The M-MΦ subtypes were fixed, permeabilized and immuno-stained for α-SMA (green) and β-tubulin (red), after a polarization time of 48 h. The cell nuclei were stained with DAPI (blue). Representative images of donor-derived M- MΦ cultures are shown. Scale bar = 25 μm. (**C**) The α-SMA-corrected total cell fluorescence in human M-MΦ subtypes (n, sample = 3; n, biological = 2; n, technical = 9) was determined by analysis of the immunofluorescence images. All data were analyzed using a one-way ANOVA, followed by Tukey’s multiple comparison tests, and expressed as mean ± SEM. ns (not significant), *p* < 0.001 (***), *p* < 0.0001 (****).

**Figure 4 biomedicines-10-02869-f004:**
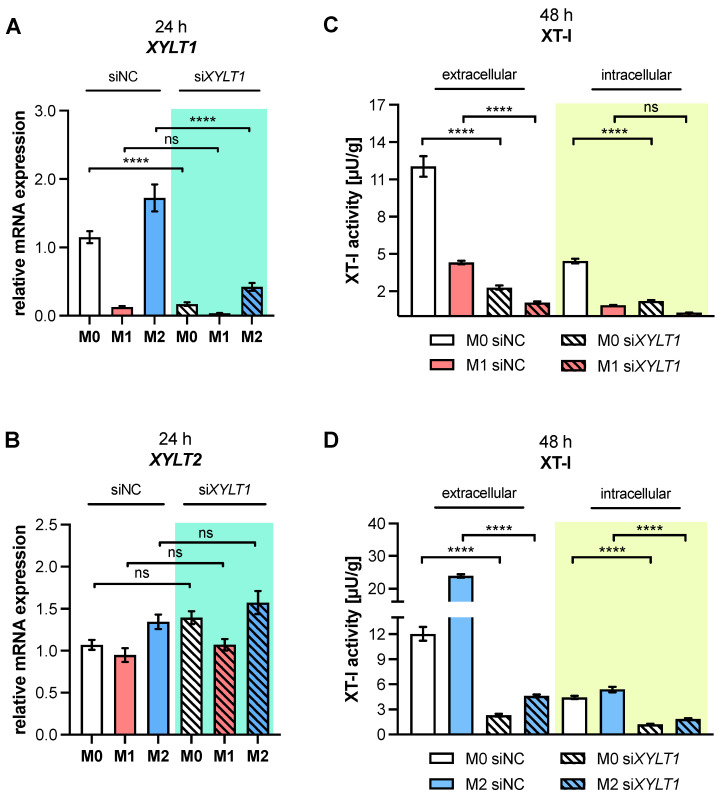
*XYLT1* mRNA expression and cellular XT-I activity of polarized macrophages after siRNA-mediated *XYLT1* knockdown. Negatively selected monocytes were differentiated into macrophages using M-CSF. On day 5, MΦ were treated with a non-targeting negative control siRNA (siNC) or an siRNA targeting *XYLT1* (si*XYLT1*). On day 6, MΦ were stimulated with IFN-γ/LPS (M1), IL-4 (M2), or with no additive (M0). Cells were harvested for qRT-PCR analysis (n, sample = 3; n, biological = 2; n, technical = 3) after a polarization time of 24 h, in order to determine the relative gene expressions of (**A**) *XYLT1* and (**B**) *XYLT2* using the ΔΔC_T_ method. (**C**) M1 and (**D**) M2 M-MΦ were harvested 48 h post-transfection for cellular XT-I activity determination (n, sample = 3; n, biological = 3; n, technical = 3) of the *XYLT1* knockdown cultures via mass spectrometry. The cell culture supernatant was used for extracellular XT-I activity, while the cell lysate was used for intracellular XT-I activity measurement. All data were analyzed using one-way ANOVA, followed by Tukey’s multiple comparison tests, and expressed as mean ± SEM. ns (not significant), *p* < 0.0001 (****).

**Figure 5 biomedicines-10-02869-f005:**
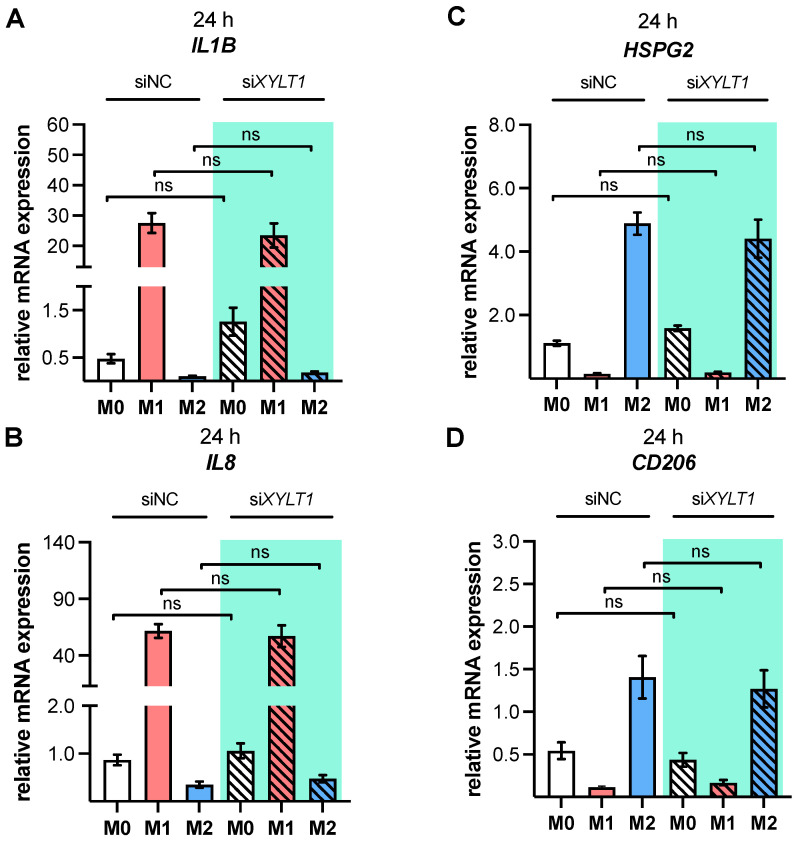
The mRNA expressions of two representative M1- and M2-like polarization state markers after siRNA-mediated *XYLT1* knockdown of polarized M-MΦ. Monocytes were differentiated into macrophages using M-CSF. On day 5, M-MΦ were treated with a non-targeting negative control siRNA (siNC) or an siRNA targeting *XYLT1* (si*XYLT1*). On day 6, M-MΦ were stimulated with IFN-γ/LPS (M1), IL-4 (M2), or with no additive (M0). Cells were harvested after a polarization time of 24 h, in order to determine the relative gene expressions of (**A**) *IL1B*, (**B**) *IL8*, (**C**) *HSPG2* and (**D**) *CD206*, via qRT-PCR using the ΔΔC_T_ method. The data (n, sample = 3; n, biological = 2; n, technical = 3) were analyzed using a one-way ANOVA, followed by Tukey’s multiple comparison tests, and expressed as mean ± SEM. ns (not significant).

**Figure 6 biomedicines-10-02869-f006:**
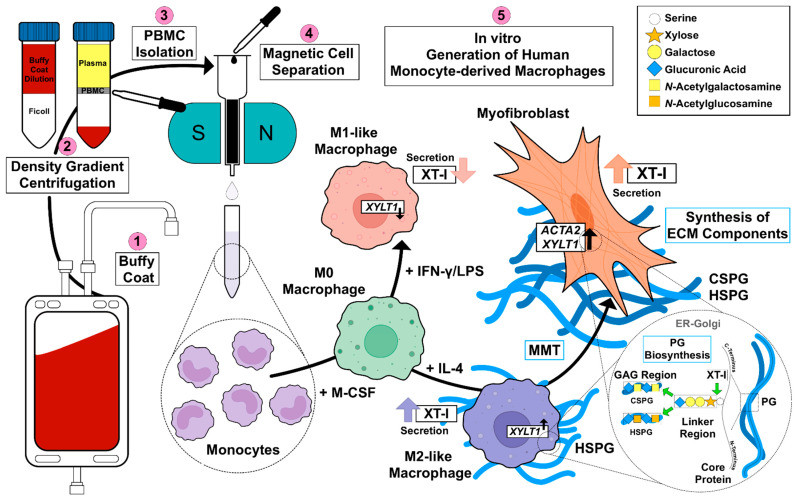
Schematic illustration of the main findings of this study, and the relationships between PBMC, MΦ subtypes, XT-I, PG and GAG.

**Table 1 biomedicines-10-02869-t001:** Summary of the protein marker expressions used for phenotypic characterization of the MΦ subtypes generated in this study. The maturation of MΦ was performed using M-CSF (M) or GM-CSF (GM), while activation was performed with IFN-γ/LPS (M1), IL-4 (M2), or with no additive (M0). Data were analyzed with a one-way ANOVA, followed by Tukey’s multiple comparison tests: ns (not significant), *p* < 0.05 (*), *p* < 0.01 (**), *p* < 0.001 (***), *p* < 0.0001 (****). The comparisons shown refer to the latter MΦ subtype. Significant relative expression increases are indicated by the symbol “▲”, and decreases by the symbol “▼”. Nonsignificant increases or reductions are indicated by the symbols “▲” and “▼”, respectively, while changes in no particular direction are indicated by “↔”.

MΦ Subtypes	HLA-DR	CD14	CD16	CD163	CD80	CD206	IL-6
**M0 M** vs. **M1 M**	▲ (*)	▼ (****)	▼ (****)	▼ (*)	▲ (****)	▼ (ns)	▲ (*)
**M0 M** vs. **M2 M**	▲ (ns)	▼ (****)	▼ (*)	▼ (ns)	▲ (ns)	▲ (**)	↔ (ns)
**M1 M** vs. **M2 M**	▼ (ns)	▲ (**)	▲ (**)	▲ (ns)	▼ (***)	▲ (***)	▼ (*)
**M0 GM** vs. **M1 GM**	▲ (*)	▼ (ns)	▼ (ns)	▼ (ns)	▲ (*)	▼ (*)	▲ (ns)

**Table 2 biomedicines-10-02869-t002:** Summary of the results from the genotypic characterization of the MΦ subtypes generated in this study. MΦ generation was performed using M-CSF (M) or GM-CSF (GM), while activation was performed with IFN-γ/LPS (M1), IL-4 (M2), or with no additive (M0). Data were analyzed using one-way ANOVA, followed by Tukey’s multiple comparison tests: ns (not significant), *p* < 0.05 (*), *p* < 0.01 (**), *p* < 0.001 (***), *p* < 0.0001 (****). Comparisons shown refer to the latter MΦ subtype. Significant relative expression increases are indicated by the symbol “▲”, and decreases by the symbol “▼”. Nonsignificant increases or reductions are indicated by the symbols “▲” and “▼”, respectively.

MΦ Subtypes	*IL1B*	*IL8*	*HSPG2*	*SDC2*
**M0 M** vs. **M1 M**	▲ (*)	▲ (**)	▼ (ns)	▼ (**)
**M0 M** vs. **M2 M**	▼ (ns)	▼ (ns)	▲ (****)	▲ (ns)
**M1 M** vs. **M2 M**	▼ (***)	▼ (**)	▲ (****)	▲ (****)
**M0 GM** vs. **M1 GM**	▲ (****)	▲ (****)	▼ (****)	▼ (***)

## Data Availability

Not applicable.

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
