# Peer review of "The Human Myofibroblast Marker Xylosyltransferase-I: A New Indicator for Macrophage Polarization"

_biomedicines, 2022, doi:10.3390/biomedicines10112869_

Round 1
Reviewer 1 Report
In the manuscript, “New Role of Myofibroblast Marker Xylosyltransferase-I in Macrophage Polarization,” the authors used an in vitro model of primary human monocyte-derived macrophages to evaluate the expression and role of xylosyltransferase-I (XT-I) in M1-like and M2-like macrophages. The authors present an in vitro model of macrophage maturation and polarization from human peripheral monocytes. The authors fully characterize their in vitro model and provide convincing evidence that their model is consistent with the published literature. They then show the novel finding XYLT1 expression and XT-I function is higher in M2-like macrophages. This finding potentially adds to our understanding of how this protein (XT-I) could be involved in MMT in disease. Overall, this work is clearly presented and explained. On occasion, the authors over-interpret results more strongly than perhaps is warranted from the data (an example is Fig 5 ACTA2 data, see below). The major weakness, however, of this work is its reliance on one in vitro model of macrophage biology. There is clearly a benefit from looking at XT-I in human tissue directly, but the tools allowing for mechanistic interpretation are limited. The paper would be greatly enhanced by follow up studies in animal models where the effect of XT-1 can be directly interrogated (inducible cell-targeted knock out of XLT1), or examined in in tissue-specific macrophages (particularly skin and lung, since authors are interested in SSc) in various mouse models. See below for additional comments.
Major comments:
Table A1 is unclear. The “regulation” column is meant to depict the expected change as the Mac is polarized toward the noted M1 or M2 phenotype? Or something else? Since the authors are including this to make interpretation easier, additional clarity on this table would help, especially in terms of confirming the authors assertion that their in vitro model is in agreement with the literature. Having table A1 and A2 be directly comparable (ie changes expected M0 vs M1 vs M2) would be helpful.
The manner in which the authors choose to split the data characterizing their model between figures and appendix figures is difficult on the reader. It is unclear why some data is in the appendix, especially as the first several figures discussed in the results section are in the appendix vs body of paper. It would be most helpful to have the tables in the main body of the text. Re-organizing this section, and possibly adding the tables to the main figure could be helpful. Alternatively, just presenting the tables and having all the surface marker data in the appendix is another possibility. Since this manuscript is largely about XT-I and not characterization of polarized macs, this reorganization could put more emphasis on the novel data presented here instead of model validation.
Figure A1A, HLA-DR. The authors posit that this marker can distinguish M1 from M2 polarization. However, as presented there is not significant difference between M1 and M2 MFI, and the authors did not polarize toward M2 after GM-CSF treatment. Based on the data, they can only really conclude (as they did in the text) that HLA-DR is different between M0 and M1.
How do the authors explain the difference in what they see for CD14/CD16 (Fig A1B&C, decreased) compared with what is predicted from literature (increased – table A1).
In fig 4, the authors only show a correlation between TGFB1 levels and XYLT1 expression and XT-1 function. Blocking TGFB or SMAD7 and examining XYLT1 levels would be demonstrate a more convincing link
The authors are over-interpreting the ACTA2 data in Fig 5, since statistically there are no significant changes between macrophage polarization types. By extension, this may weaken the interpretation of MMT somewhat.
Did the authors also look at surface marker expression in the siRNA experiments. Since the authors mentioned the markers listed in table A1, it could be expected that they examine surface marker data in addition to cytokine production. In addition, the authors only examine RNA levels, rather than surface marker protein or secreted proteins.
Minor points
Define HSPG first time it is used (line 88)
Line 43 – should be and/or
The manuscript is overall quite verbose. Some editing to shorten and streamline manuscript would improve readability.
Consider expressing data as bars with individual data points, rather than solid bars with error. This adds transparency to the data and variability.
I appreciate the detail in the Data Analysis section where sample, biological and technical replicates are explicitly defined.
Reviewer 2 Report
In the manuscript, the author utilized an in vitro model with primary human CD14+CD16+ monocyte-derived macrophages (MΦ) to investigate the role of macrophage polarization on XT-I regulation. The results identified XT-I as a novel macrophage polarization marker for in vitro generated M1- and M2-like MΦ subtypes and broadened the view of XT-I as a myofibroblast marker in the process of MMT. The manuscript is well organized and could be accepted after following questions be addressed.
1. A schematic diagram should be added to better illustrate the relationship between the Mo, M1, M2, PBMC, PG, GAG, HSPG and XT-Ι.
2. The title of the paper is "a new role of myofibroblast marker XT-1 in macrophage polarization", but the results of the paper only confirmed the difference in the expression content of XT-1 in macrophages with different phenotypes, and its new role has not been elucidated. The title should be modified.
3. The authors showed that interference with XT-1 expression did not affect macrophage phenotype, which may be related to GAG content, but the relationship has not been clarified, and how GAG changes macrophage polarization has not been mentioned. The authors should explain it.
4. The authors showed that the interference of XT-1 expression did not affect the macrophage phenotype, indicating that XT-1 may not be one of the key factors of macrophage polarization and MMT. Therefore, the value of XT-1 as a marker of macrophage polarization status needs to be confirmed.
Reviewer 3 Report
Ly et al present an interesting work illustrating the way to generate primary human macrophages and their role in the modulation of fibrotic process.
The phenotypic macrophages characterization obtained by their in vitro primary culture is very deeply described. The article is very difficult to read just because is too long.
The authors should try to reduce the introduction expecially the first part (lines 32-42). please delete line 72-73
Round 2
Reviewer 1 Report
All concerns have been adequately addressed by the authors.